# Activin A more prominently regulates muscle mass in primates than does GDF8

Esther Latres[1,*,†], Jason Mastaitis[1,*], Wen Fury[1], Lawrence Miloscio[1], Jesus Trejos[1], Jeffrey Pangilinan[1], Haruka Okamoto[1], Katie Cavino[1], Erqian Na[1], Angelos Papatheodorou[1], Tobias Willer[1], Yu Bai[1], Jee Hae Kim[1], Ashique Rafique[1], Stephen Jaspers[1], Trevor Stitt[1], Andrew J. Murphy[1], George D. Yancopoulos[1] & Jesper Gromada[1]

Growth and differentiation factor 8 (GDF8) is a TGF-β superfamily member, and negative regulator of skeletal muscle mass. GDF8 inhibition results in prominent muscle growth in mice, but less impressive hypertrophy in primates, including man. Broad TGF-β inhibition suggests another family member negatively regulates muscle mass, and its blockade enhances muscle growth seen with GDF8-specific inhibition. Here we show that activin A is the long-sought second negative muscle regulator. Activin A specific inhibition, on top of GDF8 inhibition, leads to pronounced muscle hypertrophy and force production in mice and monkeys. Inhibition of these two ligands mimics the hypertrophy seen with broad TGF-β blockers, while avoiding the adverse effects due to inhibition of multiple family members. Altogether, we identify activin A as a second negative regulator of muscle mass, and suggest that inhibition of both ligands provides a preferred therapeutic approach, which maximizes the benefit:risk ratio for muscle diseases in man.

[1] Regeneron Pharmaceuticals, Inc., 777 Old Saw Mill River Road, Tarrytown, New York 10591, USA. * These authors contributed equally to this work. † Present address: JDRF International, New York, New York 10004, USA. Correspondence and requests for materials should be addressed to J.G. (email: jesper.gromada@regeneron.com).

GDF8, also known as myostatin, is a member of the TGF-β superfamily, which acts as a negative regulator of muscle mass[1,2]. Several genetic and biochemical studies have shown that inhibition of GDF8 promotes skeletal muscle hypertrophy[3–5]. There have been considerable efforts to develop therapeutics that antagonize GDF8 signalling for treating conditions associated with loss of muscle mass and strength in humans[4–10]. However, the muscle hypertrophy induced by GDF8 inhibition has not been as effective in humans as in mice. In part, this may be because GDF8 is not the only negative regulator of muscle mass acting via the activin receptor type IIB (ACVR2B; ActRIIB). A soluble form of ActRIIB fused to human IgG Fc fragment (ActRIIB.hFc) increased muscle growth in GDF8-deficient ($Mstn^{-/-}$) mice[11], suggesting the presence of additional negative regulators of muscle hypertrophy. ActRIIB binds numerous ligands, including the activins (activin A, B, C and E), GDF11, bone morphogenetic protein 9 (BMP9) and BMP10 (refs 4,5,12). Due to its pronounced efficacy on skeletal muscle mass in animal models, ActRIIB.hFc progressed into clinical testing where it significantly increased lean body mass to a greater degree than GDF8-specific inhibition[10,13]. However, clinical trials were stopped due to bleeding complications in boys with Duchenne muscular dystrophy[14]; this effect was proposed to result from blocking BMP9, since this ligand has been linked to endothelial cell function[15,16]. Because of the numerous ligands acting via ActRIIB, therapeutic application of its inhibition seems likely to be limited by the possibility of adverse events in humans. Identification of specific ligands with inhibitory functions on skeletal muscle could allow for increased muscle growth in the absence of the adverse effects resulting from broader ActRIIB inhibition.

Evidence for the involvement of activins in regulating skeletal muscle mass came from a study which compared the efficacy of follistatin, a secreted protein that binds GDF8, GDF11 and the activins, and a follistatin mutant which binds GDF8 and GDF11 but fails to bind the activins, in increasing muscle mass[17]. Both follistatin variants induced muscle hypertrophy in mice, although the mutant lacking activin binding activity was less effective. In addition, follistatin, but not the mutant follistatin, was able to induce muscle hypertrophy in $Mstn^{-/-}$ mice, suggesting that blockade of the activins, but not of GDF11, could contribute to muscle hypertrophy in the absence of GDF8 (ref. 11). Evidence specifically implicating activin A, as opposed to activin B or C or E, came from the observation that mice deficient in one allele encoding activin A ($Inhba^{+/-}$), but not deficient in activin B ($Inhbb^{-/-}$) or activin C and E ($Inhbc^{-/-}$; $Inhbe^{-/-}$) have increased muscle mass[18]. These data suggest activin A might act in concert with GDF8 to regulate skeletal muscle mass.

To test the possibility that activin A might be a key negative regulator of skeletal muscle growth that, together with GDF8, could account for the effects of ActRIIB.hFc on muscle hypertrophy, we developed a fully human monoclonal antibody, REGN2477, that inhibits activin A, but does not bind other TGF-β family members. In addition, we used REGN2477 in combination with REGN1033, an anti-GDF8 monoclonal antibody[7]. We show that antibody-mediated inhibition of GDF8 and activin A leads to a strong and synergistic increase in muscle mass in mice and monkeys similar to that induced by treatment with ActRIIB.hFc. These results provide the first direct evidence that activin A plays a critical role for adult muscle mass maintenance and growth, and that it has a more significant role in regulating muscle mass in primates than in mice. The data provide support for combined GDF8 and activin A antibody inhibition for the treatment of muscle atrophy in patients with wasting disorders of muscle.

## Results

### In vitro characterization of activin A antibody REGN2477.

REGN2477 has high affinity ($K_D = 6$ pM) and specificity to activin A (Supplementary Table 1). This applies to mouse, monkey and human since activin A is conserved across these species. In a bioassay using A204 cells which stably express Smad-dependent luciferase driven by CAGA12 promoter (developed at Regeneron)[7], activin A, activin B, activin AB, activin AC, GDF8 and GDF11 stimulated Smad2/3 signalling (Supplementary Fig. 1). REGN2477 blocked receptor activation by a constant concentration of activin A with a half-maximal inhibitory concentration ($IC_{50}$) of 26 pM. REGN2477 also blocked receptor activation induced by activin AB ($IC_{50} = 14$ nM) and activin AC ($IC_{50} = 7.3$ nM), but not by activin B, GDF8 or GDF11 (Supplementary Fig. 1; Supplementary Table 2). ActRIIB.hFc is well known to interact with numerous TGF-β family ligands and therefore blocked receptor activation by all five tested ligands (Supplementary Fig. 1; Supplementary Tables 1 and 2).

### GDF8 and activin A inhibition increases muscle mass in mice.

Subcutaneous administration of ActRIIB.hFc to 9-week-old male SCID mice for 21 days increased tibialis anterior (TA) muscle weight by 21.9 mg or 48.8% ($n = 7$) (Fig. 1a; Supplementary Fig. 2b). The half-maximal effective concentration ($EC_{50}$) of ActRIIB.hFc was 3.0 mg kg$^{-1}$. REGN1033 increased TA muscle weight by 8.0 mg or 20.1% ($n = 5$–10) (Fig. 1b; Supplementary Fig. 2f). The $EC_{50}$ for REGN1033 was 0.52 mg kg$^{-1}$. REGN2477 (10 mg kg$^{-1}$) produced a small (1.4 mg or 5.0%; $n = 7$) and non-significant increase in TA muscle weight (Fig. 1c; Supplementary Fig. 2j, dotted line). A similar small increase was observed following dosing of 25 mg kg$^{-1}$ REGN2477 (1.1 mg or 2.5%; $n = 5$; Supplementary Fig. 3). However, administration of increasing doses of REGN1033 in the presence of 10 mg kg$^{-1}$ REGN2477 caused dose-dependent and pronounced increases in TA muscle weight (Fig. 1c). The effects were synergistic and amounted to 20.8 mg or 43.9% ($n = 7$) increase in TA muscle mass over the 21-day dosing period (Fig. 1c; Supplementary Fig. 2j). The $EC_{50}$ was 1.4 mg kg$^{-1}$ REGN1033. Similar maximal increases in TA muscle weight were observed (21.5 mg or 49.8%; $n = 5$–6) when increasing doses of REGN2477 were administrated to mice receiving a maximally effective dose of REGN1033 (10 mg kg$^{-1}$; Fig. 1d and Supplementary Fig. 2n). The $EC_{50}$ was 0.4 mg kg$^{-1}$ REGN2477. The changes in body weight and gastrocnemius (GA) muscle weights at the different treatment conditions are shown in Supplementary Fig. 2. The increases in TA muscle mass in mice treated with REGN1033, combination of REGN1033 and REGN2477, or ActRIIB.hFc resulted from hypertrophy as revealed by larger fibre cross-sectional areas rather than from an increase in fibre number (Fig. 1e–g). The induction of muscle hypertrophy was not associated with changes in the expression of $Mstn$ or $Inhba$ (Fig. 1h,i) or circulating GDF8 or activin A levels (Fig. 1j,k). As expected, the antibodies showed strong target engagement resulting in 45-fold higher total GDF8 levels in plasma with REGN1033 and 15-fold higher total activin A levels in plasma the presence of REGN2477 (Fig. 1j,k). It was not possible to measure plasma GDF8 and activin A levels in the presence of ActRIIB.hFc since it interfered with the assays. The ability of REGN1033 and REGN2477 to induce muscle hypertrophy was not restricted to SCID mice as comparable increases in muscle mass were observed in C57BL/6 mice (Fig. 1l). Given that GDF11 is closely related to GDF8 and also a ligand for ActRIIB (ref. 19), we tested if antibody blockade of GDF11 would further increase muscle hypertrophy over that seen with REGN1033 and REGN2477. To that end, we used a high-

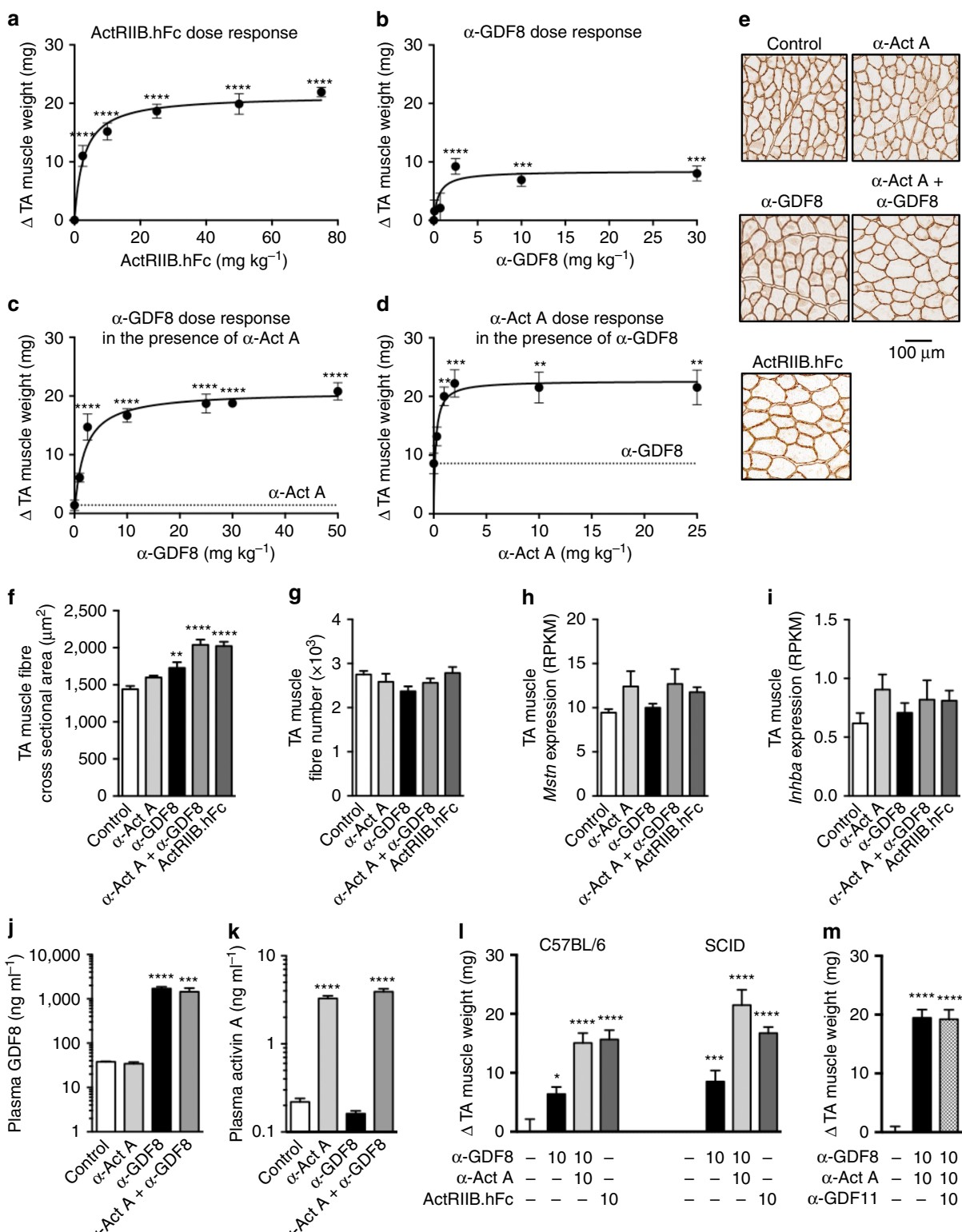

affinity ($K_D = 7.80E-11$ M) and specific α-GDF11 blocking antibody (Supplementary Fig. 3). Administration of the α-GDF11 antibody to 9-week-old male SCID mice in the presence of REGN1033 and REGN2477 (all at 10 mg kg$^{-1}$) for 21 days did not further increase muscle hypertrophy over that observed with the REGN1033 and REGN2477 combination (Fig. 1m). This is not a surprising finding since circulating levels of GDF11 are very low in mice (cf. Fig. 5d and ref. 20). In summary, these data show that simultaneous antibody inhibition of GDF8 and activin A caused a large and synergistic increase in muscle mass in mice resulting from muscle hypertrophy and that the magnitude of these effects were similar to those observed for ActRIIB.hFc. This

demonstrates that activin A acts to restrain skeletal muscle hypertrophy in mice.

**GDF8 and activin A inhibition increases muscle force in mice.** Nine-week-old male SCID mice were treated with REGN1033 or REGN2477 alone or in combination as well as with ActRIIB.hFc

at 10 mg kg$^{-1}$. At the end of the 21-day dosing period, TA muscles were isolated for *ex vivo* isometric force measurements. Consistent with our previous report[7], we found that REGN1033 increased muscle twitch force by 12% (Fig. 2a). Muscle from mice treated with the combination of REGN2477 and REGN1033 showed a larger increase in twitch force (33%) than expected from the sum of effects of REGN2477 (9%) and REGN1033

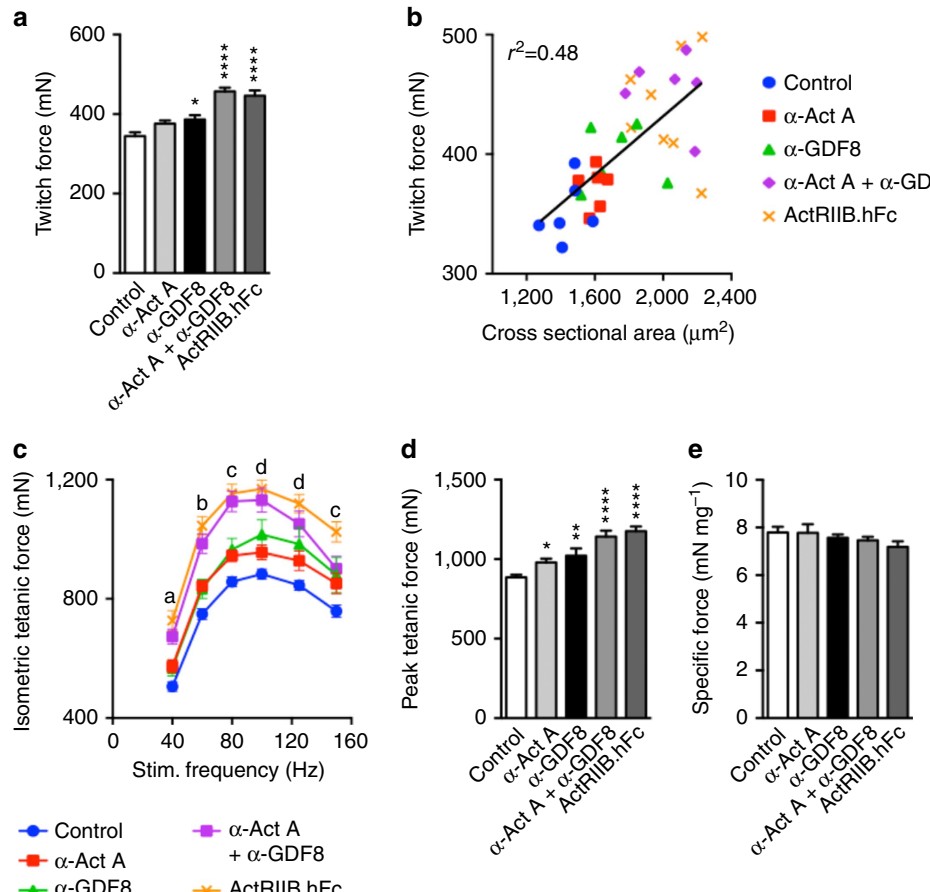

**Figure 2 | Activin A and GDF8 inhibition increases muscle force in mice.** (**a**) Twitch force of TA muscle from mice injected with 10 mg kg$^{-1}$ α-Act A (*n* = 15), α-GDF8 (*n* = 10), the combination of α-Act A and α-GDF8 (*n* = 9), ActRIIB.hFc (*n* = 10) or isotype control antibody (*n* = 22) for 21 days. (**b**) Correlation of twitch force to fibre cross-sectional area for a subset of the mice described in **a**. (**c**) Isometric tetanic force in *ex vivo* TA muscle over stimulation frequencies of 40–150 Hz for the mice described in **a**. (**d**) Peak tetanic force from the groups in **c**. (**e**) Specific force for the groups in **c**. Data are shown as mean ± s.e.m. *$P < 0.05$, **$P < 0.01$, ****$P < 0.0001$ versus control. For **c**, a = α-Act A + α-GDF8 and ActRIIB.hFc groups significant to control, b = α-Act A, α-ActA + α-GDF8 and ActRIIB.hFc groups significant to control, c = α-Act A, α-GDF8, α-Act A + α-GDF8 and ActRIIB.hFc groups significant to control, d = α-GDF8, α-Act A + α-GDF8 and ActRIIB.hFc groups significant to control. Statistical significance was calculated using one-way (**a**,**d**) or two-way (**c**) ANOVA with Bonferroni *post hoc* test.

**Figure 1 | Activin A and GDF8 inhibition synergistically increase muscle mass in mice.** Changes in TA muscle weight following administration of increasing doses of (**a**) ActRIIB.hFc (*n* = 7), (**b**) α-GDF8 (*n* = 10, except for 0.1 and 0.75 mg kg$^{-1}$ doses, *n* = 5), (**c**) α-GDF8 in the presence of 10 mg kg$^{-1}$ α-Act A (*n* = 7, except for 2.5, 25 and 50 mg kg$^{-1}$ doses, *n* = 8) and (**d**) α-Act A in the presence of 10 mg kg$^{-1}$ of α-GDF8 (*n* = 5, except for 0.3 and 1.0 mg kg$^{-1}$ doses, *n* = 6) in male SCID mice for 21 days. Change in TA muscle weight was calculated as the difference versus isotype control group mean. (**e**) Representative images of laminin staining of TA muscle from mice injected with 10 mg kg$^{-1}$ of α-Act A, α-GDF8, the combination of α-Act A and α-GDF8 or ActRIIB.hFc for 21 days (*n* = 6, except for ActRIIB.hFc, *n* = 9). Mean fibre cross-sectional area (**f**) and fibre number per section (**g**) from studies described in **e**. mRNA levels of *Mstn* (**h**) and *Inhba* (**i**) from TA muscle of SCID mice treated with α-Act A or α-GDF8, the combination of the antibodies or ActRIIB.hFc for 21 days (10 mg kg$^{-1}$ each, *n* = 7). Data expressed in reads per kilobase of transcript per million mapped reads (RPKM). ELISA data for GDF8 (**j**) and activin A (**k**) from SCID mice treated for 21 days with α-GDF8, α-Act A or the combination (control group in **j**, *n* = 5, all others *n* = 6). (**l**) Change in TA muscle weights in C57BL/6 mice compared to SCID mice after 21 days of treatment (*n* = 5 for C57BL/6 study; *n* = 12 for the control, *n* = 5 for the α-GDF8 and combination groups, *n* = 10 for the ActRIIB.hFc in SCID study). (**m**) Change in TA muscle weights in SCID mice following treatment with α-Act A and α-GDF8 combination (10 mg kg$^{-1}$ each, *n* = 7) in the absence or presence of α-GDF11 antibody (10 mg kg$^{-1}$, *n* = 6). Data are mean ± s.e.m. *$P < 0.05$, **$P < 0.01$, ***$P < 0.001$, ****$P < 0.0001$ versus control (**a**, **b**, **f**, **j–m**), 10 mg kg$^{-1}$ α-Act A (**c**) or 10 mg kg$^{-1}$ α-GDF8 (**d**) by one-way ANOVA and Bonferroni *post hoc* test.

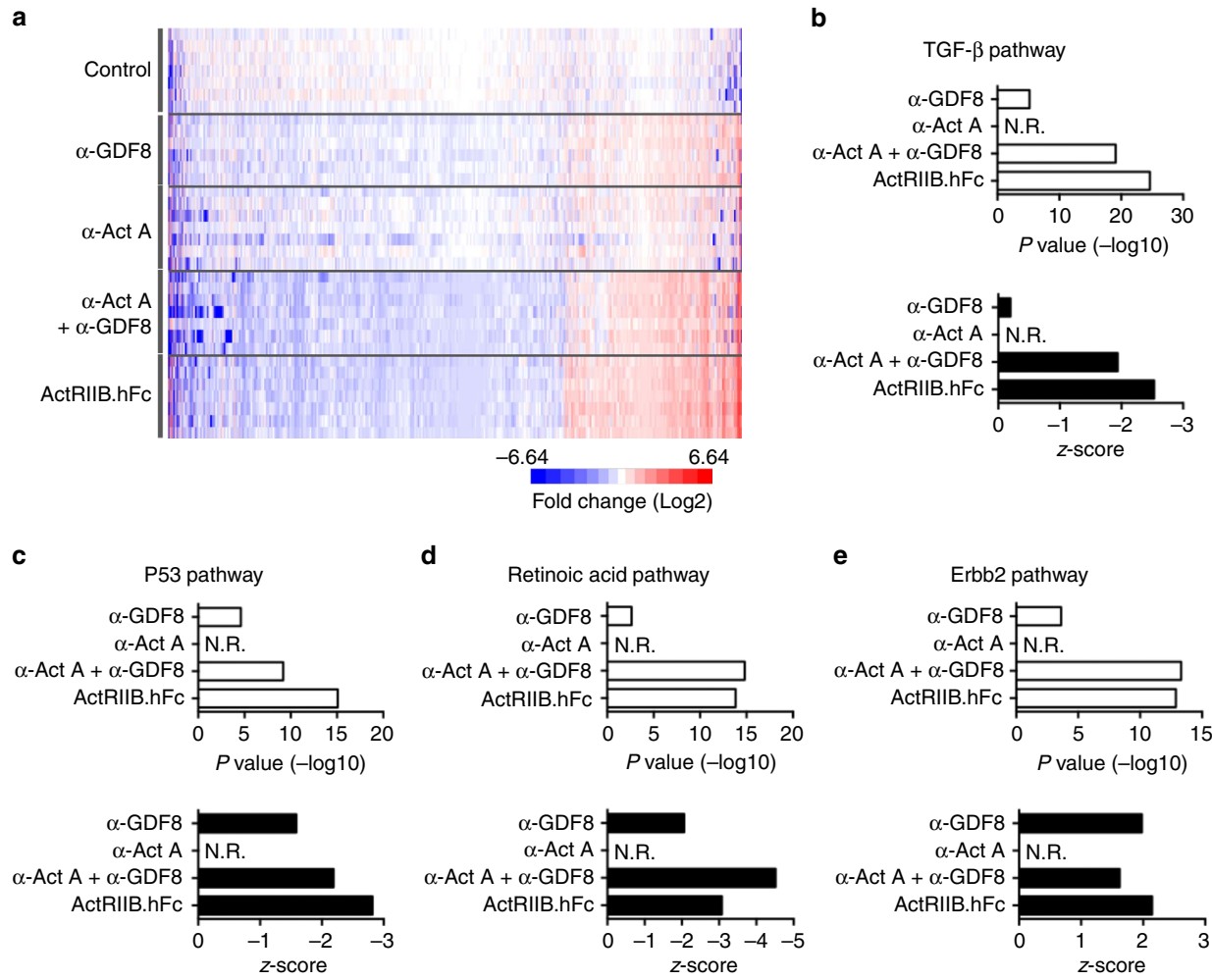

**Figure 3 | Downregulation of TGF-β pathway genes in TA muscle by RNAseq.** (**a**) Heat map of the union of 1,670 genes perturbed by α-GDF8, α-Act A, the combination of α-Act A and α-GDF8 or ActRIIB.hFc following dosing at 10 mg kg$^{-1}$ for 10 days (n = 7). (**b**) IPA analysis of TGF-β pathway as measured by P value and z-score from each treatment group. (**c–e**) IPA analysis showing P value and z-score of other top-regulated common pathways in the α-Act A and α-GDF8 combination and ActRIIB.hFc treatment groups. N.R., not regulated.

(12%). The increase in force production was similar to that observed with ActRIIB.hFc (30%) (Fig. 2a). We found a correlation between fibre cross-sectional area and twitch force (Fig. 2b). REGN1033 increased isometric force production at all stimulation frequencies (Fig. 2c). REGN2477 caused a small increase in force production. The combination of REGN1033 and REGN2477 as well as ActRIIB.hFc produced larger increases in isometric force production (Fig. 2c). We observed a similar pattern of effects on peak tetanic force (Fig. 2d). No change in specific force was observed for any of the treatment groups (Fig. 2e). These data show that the large increase in muscle mass following antibody inhibition of GDF8 and activin A translated into greater muscle force, an effect comparable to that observed with ActRIIB.hFc.

**Reduced TGF-β pathway activity in mouse skeletal muscle.** We analysed mRNA levels of genes expressed in TA muscle from mice treated with REGN1033 or REGN2477 alone or in combination, as well as with ActRIIB.hFc at 10 mg kg$^{-1}$ for 10 days. Expression levels of affected genes are provided in Supplementary Dataset 1. The heat map of the union of perturbed genes (n = 1,670) show more pronounced gene expression changes for the combination of REGN1033 and REGN2477 than REGN1033

or REGN2477 alone, an effect that was comparable to that observed with ActRIIB.hFc (Fig. 3a). Ingenuity pathway (IPA) upstream analysis revealed that perturbed genes primarily clustered in the TGF-β signalling pathway (Fig. 3b). Consistent with the strong induction of muscle hypertrophy, we found that the activation score (z-score) was negative, indicating that the changes of the enriched genes reduced activity of the TGF-β signalling pathway (Fig. 3b). We also found that genes perturbed by the REGN1033 and REGN2477 combination and ActRIIB.hFc reduced the activity of the p53 and retinoic acid pathways (Fig. 3c,d). ERBB2 was the only upregulated pathway (Fig. 3e). ERBB2 is the receptor for neuregulin and plays a role in the formation of the neuromuscular synapses as well as in the regulation of skeletal muscle protein synthesis and metabolism[21–24]. These data show that simultaneous inhibition of GDF8 and activin A strongly reduced expression of genes in the TGF-β signalling pathway, consistent with the induction of muscle hypertrophy. Interestingly, the activity of other cell growth and differentiation pathways was also perturbed and may contribute to expansion of the muscle mass.

**GDF8 and activin A inhibition improve recovery from atrophy.** To test whether antibody blockade of GDF8 and activin A improve recovery from muscle atrophy, we exposed 9-week-old

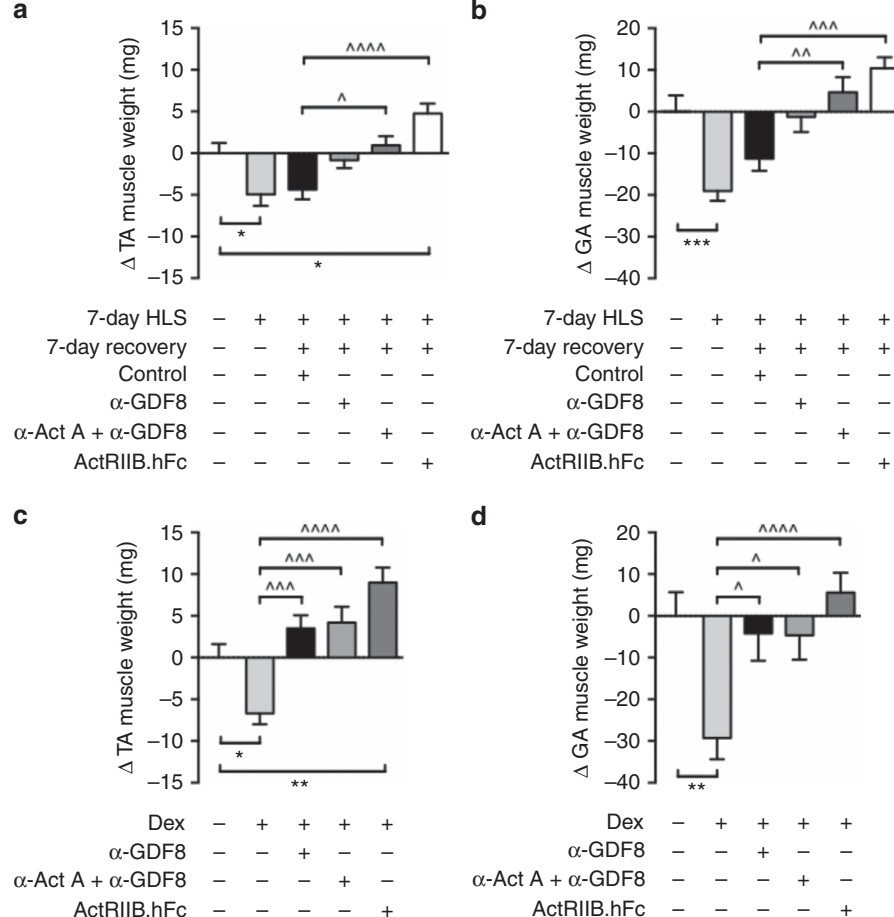

**Figure 4 | Activin A and GDF8 inhibition improve recovery from atrophy in mice.** (**a,b**) Change in TA and GA muscle weights of C57BL/6 mice following 7 days of HLS and 7 days of recovery. Immediately or 3 days after 7-day recovery, mice were treated with either an isotype control antibody (25 mg kg$^{-1}$), α-GDF8 alone or in combination with α-Act A (each 25 mg kg$^{-1}$) or ActRIIB.hFc (25 mg kg$^{-1}$). Untreated and HLS control mice were euthanized at day 7, all other groups were euthanized at day 14 of the study, $n = 10$ per group. (**c,d**) Change in TA and GA muscle weights in mice infused with dexamethasone (Dex; 23 μg per day) for 14 days and treated with α-GDF8 alone or in combination with α-Act A (each 25 mg kg$^{-1}$) or ActRIIB.hFc (25 mg kg$^{-1}$). Mice were all given four doses on D0, D3, D7 and D10. Control group was infused with saline. Data are mean ± s.e.m. (**a,b**) \*$P < 0.05$, \*\*$P < 0.01$, \*\*\*$P < 0.001$ versus no HLS. ^$P < 0.05$, ^^$P < 0.01$, ^^^$P < 0.001$, ^^^^$P < 0.0001$ versus HLS control. (**c,d**) \*$P < 0.05$, \*\*$P < 0.01$ versus saline control. ^$P < 0.05$, ^^^$P < 0.001$, ^^^^$P < 0.0001$ versus Dex control. Statistical significance was calculated by one-way ANOVA with Bonferroni *post hoc* test.

male C57BL/6 mice to 1 week of hindlimb-induced atrophy, whereby mice were suspended by tail to remove loading from their hind legs. One hindlimb suspension (HLS) group and one non-suspended control group were euthanized after 7 days. Mice in the HLS group lost 5 mg (10.4%) of TA and 19 mg (11.9%) of GA muscle weight when compared to the unsuspended control group (Fig. 4a,b). The remaining suspended mice were randomized and then treated with 25 mg kg$^{-1}$ of REGN1033 alone or in combination with REGN2477 or ActRIIB.hFc for 7 days. Little recovery of muscle mass was observed for the control antibody treated mice (Fig. 4a,b). REGN1033 significantly improved recovery from HLS and their TA and GA muscle weights were not significantly different from those in untreated control mice. The REGN1033 and REGN2477 combination not only normalized muscle weight but also induced slight muscle hypertrophy relative to the muscle weights in untreated control mice (Fig. 4a,b). The effects of the REGN1033 and REGN2477 combination were comparable or slightly smaller than the increases in TA and GA muscle weights observed in ActRIIB.hFc treated mice (Fig. 4a,b).

We also induced muscle atrophy in 9–10-week-old male C57BL/6 mice by treatment with dexamethasone (23.0 μg per day) for 14 days. Dexamethasone induced slightly more TA and

GA muscle atrophy than observed in the HLS model. REGN1033 treatment reversed dexamethasone-induced muscle atrophy and even produced a small increase in TA muscle weight over control mice (Fig. 4c,d). Surprisingly, treatment of mice with REGN1033 and REGN2477 did not produce a further increase in muscle weight over that observed with REGN1033 treatment. ActRIIB.hFc prevented dexamethasone-induced muscle atrophy, although the effects were not significantly greater than those observed with REGN1033 alone or in combination with REGN2477 (Fig. 4c,d). These data show that the combination of REGN1033 and REGN2477 facilitates recovery of muscle mass following atrophy in mice, an effect comparable to that observed with ActRIIB.hFc.

**GDF8 and activin A inhibition increases lean mass in monkeys.** GDF8 functions to control skeletal muscle growth across all mammalian species[25]. To see if the same applies to activin A, we administered REGN2477 weekly at 3, 15 or 50 mg kg$^{-1}$ for 5 weeks and saw small increases in lean body mass measured by dual x-ray absorptiometry (DXA) in cynomolgus monkeys (Fig. 5a). Administration of REGN1033 at 50 mg kg$^{-1}$ increased lean body mass to a similar extent to that observed

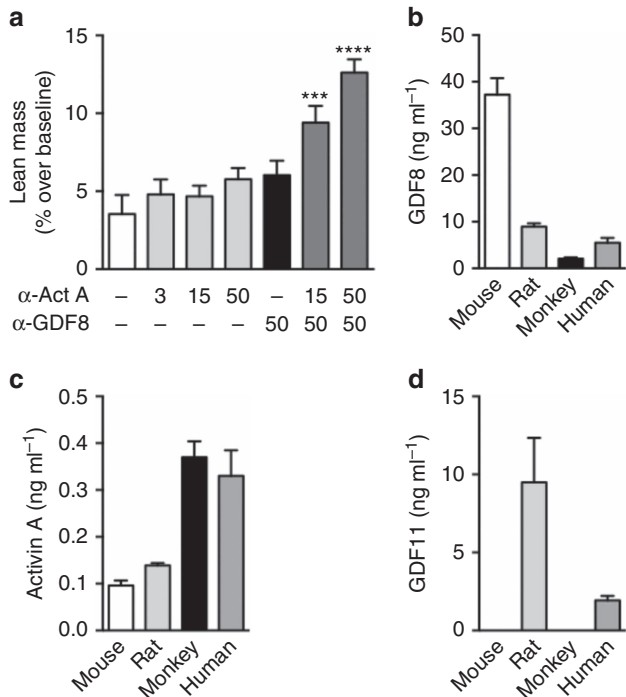

**Figure 5 | Activin A and GDF8 inhibition synergistically increase lean mass in monkeys.** (**a**) Per cent change in lean mass over baseline in male and female cynomolgus monkeys ($n = 5$ per gender per group). Doses shown in mg kg$^{-1}$. Basal serum levels of (**b**) GDF8, (**c**) activin A and (**d**) GDF11 in SCID mice ($n = 8$), rats ($n = 8$), cynomolgus monkeys ($n = 6$) and humans ($n = 5$ per gender; 18–43 years of age). Data are shown as mean ± s.e.m. \*\*\*$P < 0.001$, \*\*\*\*$P < 0.0001$ versus saline control by one-way ANOVA and Bonferroni *post hoc* test.

for REGN2477 at an equivalent dose (Fig. 5a). Interestingly, the combination of REGN2477 and REGN1033 caused a dose-dependent and synergistic increase in lean body mass. The increase was 9.1% at the maximal tested doses ($P < 0.0001$; $n = 10$). No changes in body fat mass were observed for any of the dosing groups (Supplementary Fig. 5).

The relatively small effect of GDF8 inhibition and the similar contributions of GDF8 and activin A blockade on lean body mass in monkeys prompted us to measure circulating levels of the ligands. Figure 5b shows that circulating GDF8 levels were 4- to 18-fold lower in monkeys, rat and humans than in mice. Interestingly, we found that activin A serum levels were three- to four-fold higher in cynomolgus monkeys and humans than in mice and rat (Fig. 5c). Using a specific and high-affinity GDF11 ELISA[20], we detected the highest circulating levels in rat and lower levels in humans (Fig. 5d). Plasma levels of GDF11 were below detection (0.43 ng ml$^{-1}$) in serum from mice and monkeys (Fig. 5d). These data suggest more equal contributions of GDF8 and activin A to regulation of lean body mass in monkeys and humans than mice and with little contribution from GDF11. This is supported by the observation that antibody neutralization of both ligands was required to produce a robust expansion of lean body mass in monkeys.

**GDF8 and Act A inhibition similar to ActRIIB.hFc in primates.** Figure 6a shows that 8 weeks of dosing is required to obtain a maximal increase in lean body mass with a combination of REGN1033 and REGN2477 (both at 50 mg kg$^{-1}$). The increase in lean body mass amounted to 13.7% and was similar to that observed with ActRIIB.hFc (13.6%; Fig. 6a). The corresponding

changes in body weight and body fat mass are shown in Supplementary Fig. 6a–c. We observed an increase in biceps muscle weight at the end of the study (REGN2477 and REGN1033: 67 ± 16%; ActRIIB.hFc: 53 ± 12%; Fig. 6b). The increase in muscle weight resulted from hypertrophy since fibre cross-sectional area increased by 15 ± 16% following REGN2477 and REGN1033 treatment and by 22 ± 15% with ActRIIB.hFc (Fig. 6c,d). As a consequence, fibre number per muscle area decreased accordingly (REGN2477 and REGN1033: − 10 ± 12%; ActRIIB.hFc: − 19 ± 12%) (Fig. 6e). No changes were observed in bone mineral content or density nor in thigh or tibia circumference (Supplementary Fig. 7d–g). These data show that antibody neutralization of GDF8 and activin A induced skeletal muscle fibre hypertrophy and strongly increased lean body and muscle mass in cynomolgus monkeys.

**ActRIIB.hFc side-effect profile in mice.** A recent study showed that ActRIIB.hFc exacerbated hyperglycaemia in a mouse model of type 1 diabetes within 1 week of administration[26]. We now show in mice that ActRIIB.hFc increased pancreas weight (Fig. 7a), blood glucose levels (Supplementary Table 3) and impaired glucose tolerance (Fig. 7b). These effects were not associated with changes in plasma insulin, glucagon or corticosterone or the mass of pancreas alpha and beta cells (Supplementary Table 3). Importantly, these adverse effects were not due to inhibition of activin A or GDF8, since mice co-treated with REGN1033 and REGN2477 had normal glucose tolerance and pancreas weight (Fig. 7a,b).

In addition to increasing pancreas weight, ActRIIb.hFc caused splenomegaly in mice (Fig. 7c). This is secondary to increased extramedullary hematopoiesis in the red pulp revealed by foci of erythroblastic islets (Fig. 7d). No increase in spleen weight or extramedullary hematopoiesis was observed in mice treated with REGN1033 and REGN2477 (Fig. 7c,d). The increase in extra-medullary hematopoiesis was supported by gene expression analysis demonstrating many differentially expressed genes in the spleen from mice treated with ActRIIb.hFc (Supplementary Fig. 7a). The perturbed genes were primarily involved in regulation of hemostasis (Supplementary Fig. 7b,c). Importantly, the gene expression profiles in spleens from mice treated with REGN1033 and REGN2477 were comparable to control antibody treated mice (Supplementary Fig. 6a). Thus, specific antibody blockade of GDF8 and activin A provide for similar muscle hypertrophy while avoiding the toxicities associated with broader ActRIIB inhibition.

## Discussion
It has long been known that GDF8 acts as an antagonist of muscle growth in animals and man. However, a variety of studies indicated that a second ligand might be involved in regulating skeletal muscle mass through the same receptor system as GDF8. We report the development of a high-affinity and specific human activin A neutralizing antibody. Administration of this antibody in combination with our GDF8-inhibiting antibody caused pronounced expansion of skeletal muscle and lean body mass in mice and cynomolgus monkeys. The hypertrophic effects were similar to those observed with the broad-acting ActRIIB.hFc that blocks numerous TGF-β ligands, but specific blockade of GDF8 and activin A avoided adverse effects associated with broad inhibition. In addition, the role of activin A appears more prominent in primates than in mice, which is consistent with the smaller relative response to GDF8-specific inhibition in humans as compared to mice. These data show that GDF8 and activin A are key negative regulators of muscle growth in mice and monkeys. Further, our results suggest that antibody neutralization of both GDF8 and activin A might provide an effective therapy

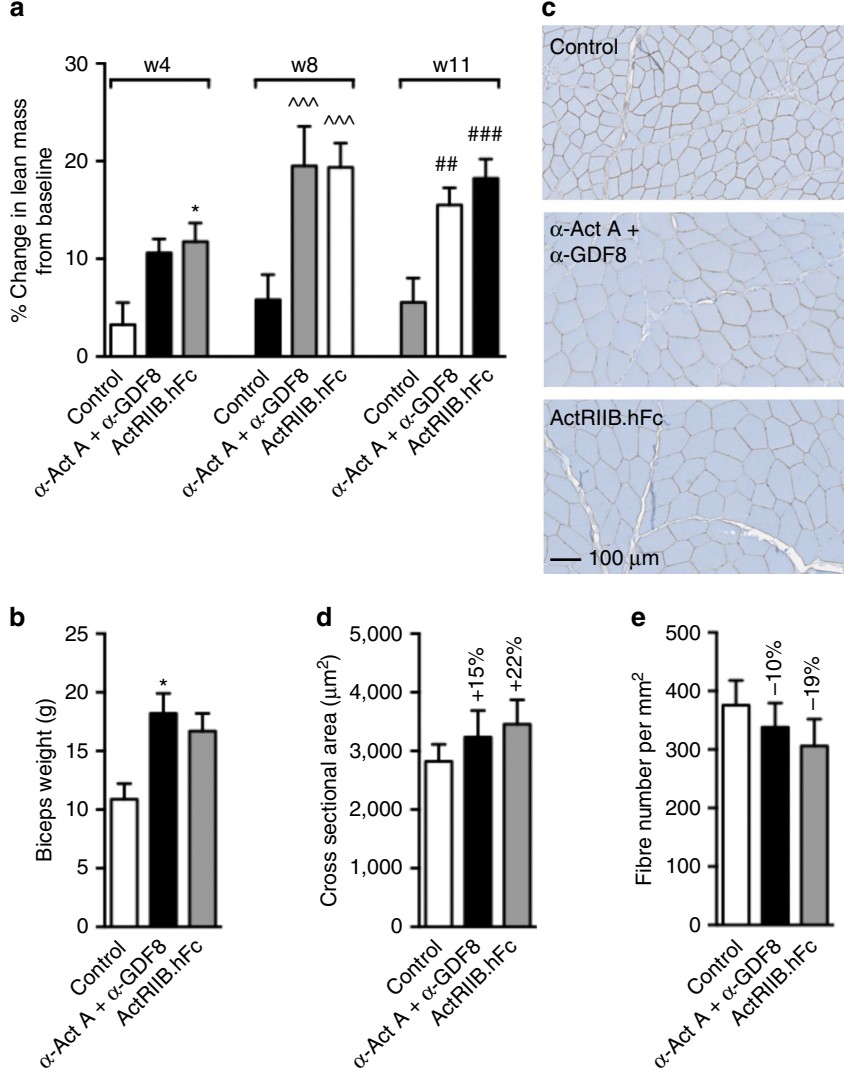

**Figure 6 | Combination treatment increases lean mass similar to ActRIIB.hFc in monkeys.** (**a**) Per cent change over baseline of lean mass in cynomolgus monkeys ($n = 3$ per gender per group) at weeks 4, 8 and 11 of dosing compared to pretreatment. Dosing was 50 mg kg$^{-1}$ of each antibody i.v. for 11 weeks. (**b**) Terminal biceps weight for each group with representative sections given in **c**. Quantitation of bicep fibre size (**d**) and fibre number (**e**) with mean values shown above bars. Data are shown as mean ± s.e.m. *$P < 0.05$ versus 4 week saline control, ^^^$P < 0.001$ versus 8 week saline control, ##$P < 0.01$, ###$P < 0.001$ versus 11 week saline control (**a**) and *$P < 0.05$ versus saline control (**b**) by one-way ANOVA with Bonferroni *post hoc* test.

for muscle atrophy and wasting diseases with an improved benefit:risk profile.

Our data show that GDF8 is the dominant negative regulator of muscle mass in mice: specific antibody neutralization of GDF8, but not that of activin A, resulted in muscle mass expansion. Since GDF8 and activin A both bind ActRIIB and activate Smad2/3 signalling with comparable potencies and efficacy, we postulate that the higher circulating GDF8 levels is the primary reason for its dominant role in controlling murine muscle mass. This explains why antibody neutralization of activin A in the absence of GDF8 antagonism caused limited muscle growth in mice. However, activin A is sufficient to suppress muscle growth in the absence of GDF8 since antibody neutralization of both ligands doubled muscle growth compared to GDF8 inhibition alone. The comparable serum levels of GDF8 and activin A in monkeys might explain why antibody neutralization of each ligand alone had little effect on muscle mass but that blocking both ligands caused a pronounced and synergistic increase in lean body mass. We speculate that this will also be the case in humans

since circulating levels of GDF8 and activin A are comparable between monkeys and humans.

One of the first observations indicating that high circulating levels of activin A cause severe muscle wasting was provided by a mouse genetic study. The biological effects of activin A are negatively regulated by inhibin A, and genetic inactivation of *Inha* resulted in elevated levels of activin A that caused a severe cachexia phenotype, including muscle wasting and decreased survival[27]. Subsequent studies have confirmed that elevated serum levels of activin A in tumour models, in response to pro-inflammatory cytokines or following overexpression induce cachexia and muscle atrophy[28–30]. In these animal models, blockade of activin A signalling with ActRIIB.hFc or modified inhibitory activin A pro-domains protected the mice from muscle wasting and increased survival[30,31]. An important role for activin A has also been established in human cancer cachexia[32] and it has been shown that serum concentrations of activin A increase with age[33]. These data suggest that REGN2477 alone might have beneficial effects on skeletal muscle mass and

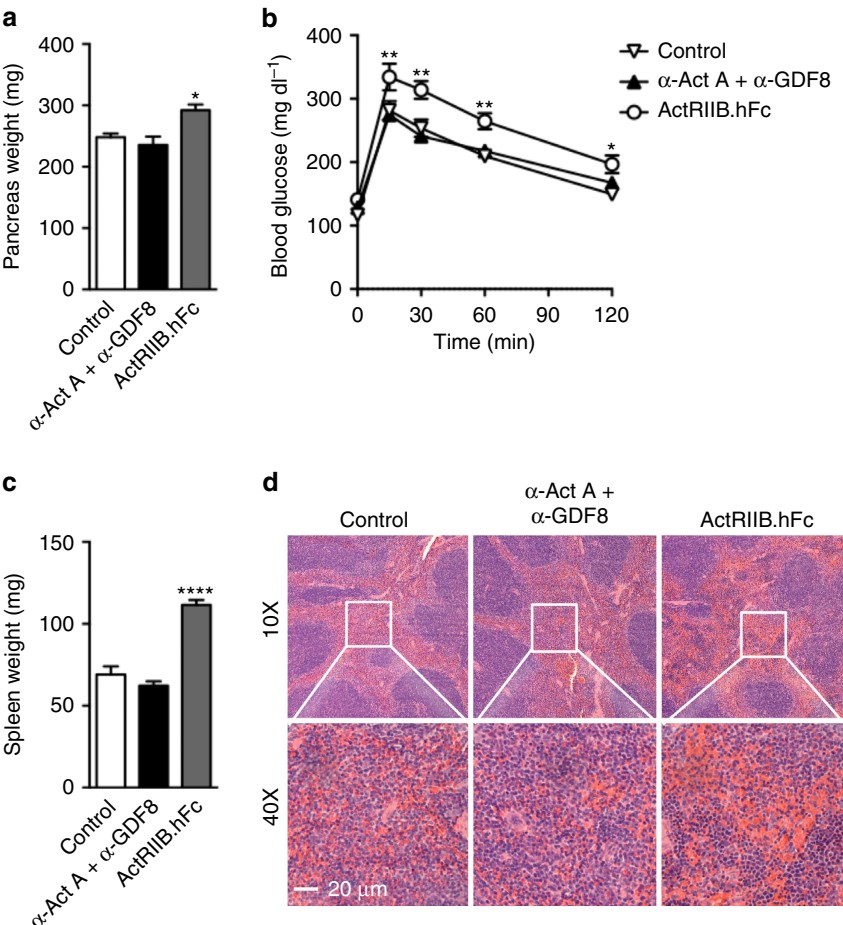

**Figure 7 | ActRIIB.hFc effects on glucose homeostasis and spleen hematopoiesis in mice.** (**a**) Pancreas weight following 8 days of treatment ($n = 6$). (**b**) Blood glucose curves following oral glucose tolerance performed after 7 days of treatment ($n = 6$). (**c**) Spleen weights following 22 days of treatment ($n = 7$). (**d**) Representative histological images of the spleen for each group following 22 days of treatment. Data are shown as mean ± s.e.m. *$P < 0.05$, **$P < 0.01$, ****$P < 0.0001$ versus control by one-way or two-way ANOVA with Bonferonni *post hoc* test.

function in conditions associated with elevated circulating activin A levels.

Our data show that GDF8 and activin A are the key regulators of muscle mass in mice and monkeys since circulating GDF11 levels are very low and that administration of a high-affinity GDF11 blocking antibody to mice already treated with GDF8 and activin A neutralizing antibodies did not further increase muscle mass. The situation is likely different in rats where we detected GDF11 levels in plasma similar to those of GDF8. Consistent with recent reports[20,34], we also detected low levels of GDF11 in human serum. Despite a few recent studies showing an age-related decrease in GDF11 and that GDF11 treatment improves muscle regeneration[35–37], it is generally accepted that both GDF8 and GDF11 induce muscle atrophy via Smad2/3 and identical downstream signalling[20,38,39]. The finding that GDF11 plasma levels decline with age in humans has also recently been contradicted[34]. This suggests that GDF11, in addition to GDF8 and activin A could modulate muscle regeneration and hypertrophy in humans.

RNA sequencing of muscle revealed that antibody blockade of GDF8 and activin A primarily perturbed genes in the TGF-β signalling pathway and reduced its activity. A similar perturbation of expressed genes in the TGF-β signalling pathway was observed with ActRIIB.hFc treatment, confirming that the actions of GDF8 and activin A are primarily mediated by ActRIIB, which signals via the ALK4/5-Smad2/3 pathway. Smad4 has been shown to act as an interface between the Smad2/3 and the BMP-Smad1/5/8 signalling pathways to promote nuclear retention and in cooperation with transcriptional co-regulators, to govern gene expression. It has been proposed that Smad4 mainly engages with the Smad2/3 pathway but switches to the BMP-Smad1/5/8 pathway when ActRIIB signalling is reduced to promote muscle hypertrophy[40]. Our gene expression analysis also revealed that reduced TGF-β-signalling activity might not be the only mechanism promoting muscle growth. Interestingly, we found that genes perturbed by the REGN1033 and REGN2477 combination and ActRIIB.hFc increased activity of the ERBB2 pathway. Neuregulin is a member of the epidermal growth factor family and can transmit signals through ERBB2. In the muscle ERBB2 is localized at the T-tubule and has been shown to regulate protein synthesis and energy metabolism[23,24,41]. ERBB2 also regulates neuromuscular synapse formation as well as muscle spindle development and survival[21,22,42]. Finally, ERBB2 acts to enhance activity of IGF1 signalling, another important regulator of skeletal muscle mass and function[43–45].

The gene expression analysis also revealed reduced activity of the tumour suppressor protein p53 pathway. p53 is a cell cycle regulator and the most frequently mutated gene in human cancers, whose inactivation leads to increased cellular growth[46]. The role of p53 in the regulation of skeletal muscle mass and function is controversial with some studies show that p53 promotes muscle cachexia and suppresses muscle

differentiation[47–49], whereas one study did not find a role for p53 in myoblast proliferation, differentiation and myotube formation *in vivo* during myogenesis of adult skeletal muscle[50]. In addition, one study showed that p53 induces myogenesis, whereas another found that p53 improves aerobic exercise capacity[51,52].

Finally, the gene expression analysis revealed reduced activity of the retinoic acid pathway in skeletal muscle from mice treated with the REGN1033 and REGN2477 combination or ActRIIB.hFc. Retinoic acid signalling is mediated through the activation of retinoic acid receptors, which function as ligand-dependent transcription factors. Retinoic acid is an important vertebrate morphogen and is essential for the correct patterning of the vertebrate embryo. Interestingly, the effects of retinoic acid are highly context dependent in the skeletal muscle, affected by locus-specific interaction or local chromatin environment[53]. For this reason, further studies are required to understand how the retinoic acid pathway interacts with other signalling pathways to promote muscle hypertrophy and improved function under conditions where GDF8 and activin A actions are inhibited. These data show that simultaneous inhibition of GDF8 and activin A strongly reduced expression of genes in the TGF-β signalling pathway, consistent with the induction of muscle hypertrophy. Interestingly, the activity of other cell growth and differentiation pathways was also perturbed and may contribute to expansion of the muscle mass.

Our data show that antibody inhibition of GDF8 and activin A mimic the effects of ActRIIB.hFc on muscle growth in mice and monkeys. This implies that GDF8 and activin A, but not other TGF-β ligands that bind ActRIIB.hFc are important for muscle mass maintenance and expansion. However, the other ActRIIB.hFc ligands have important biological functions in the body. For example, the increased risk for bleeding with ActRIIB.hFc is likely to be secondary to inhibition of BMP9 interacting with the endoglin-ALK1-BMPRII/ActRIIB signalling pathway[16,54,55]. Likewise, the acute impairment of glucose control and expansion of pancreas is also mediated by ActRIIB ligands other than activin A and GDF8. Finally, and consistent with a previous report[56], we found that spleens from mice treated with ActRIIB.hFc exhibited splenomegaly and extramedullary hematopoiesis in the red pulp. The erythropoietic growth factors BMP2 or 7 are unlikely to account for the induction of extramedullary hematopoiesis, since they are ligands for ActRIIB.hFc. A similar situation exists for BMP4, another ActRIIB.hFc ligand that has previously been shown to induce extramedullary hematopoiesis in mice[57]. Interestingly, the type III TGF-β receptor has been shown to be a marker that distinguishes 'early' and 'late' burst-forming unit erythroid progenitors[58]. 'Early' burst-forming unit erythroid progenitors have little type III TGF-β receptor expression and show maximal capacity for self-renewal and contribution to total erythroblast production[59]. Furthermore, signalling by the type III TGF-β receptor promotes transition from 'early' to 'late' burst-forming unit erythroid progenitors. Consistent with this, inhibition of TGF-β signalling increases number of 'early' burst-forming unit erythroid progenitors and total erythroblast production[59]. We speculate that ActRIIB.hFc blockade of type III TGF-β receptor ligands might increase the number of 'early' burst-forming unit erythroid progenitors in the spleen and promote extramedullary hematopoiesis. Thus, the use of ActRIIB.hFc or antibody inhibition of ActRIIB[60] as therapeutic strategies for promoting muscle growth could be associated with unwanted side effects. This implies that specific antibody neutralization of GDF8 and activin A might provide a more selective approach to profoundly increase muscle mass and strength in muscle wasting diseases.

Taken together, our data provide evidence that GDF8 and activin A are key negative regulators of muscle growth and function and that combined antibody inhibition of these two ligands promoted a strong increase in muscle and lean body mass in mice and monkeys, with a superior therapeutic window as compared to broad TGF-β ligand inhibition. These findings support the use of the GDF8-neutralizing antibody, REGN1033, and the activin A neutralizing antibody, REGN2477, for muscle atrophy and wasting diseases.

## Methods

**Antibodies and protein reagents.** REGN2477 was created by immunizing Regeneron's VelocImmune mice[61,62] with mature human activin A. The resulting antibody contains an IgG4 constant region and is a fully human monoclonal specific to activin A. The α-GDF8 blocking antibody REGN1033 was developed by crossing Regeneron's Velocimmune mice with mice in which the *Mstn* gene had been deleted. The progeny were then immunized with human GDF8 protein. The resulting antibody is a fully human IgG4 monoclonal specific for GDF8 with no cross-reactivity to GDF11. An IgG4 isotype antibody was used as control for all antibody experiments. Activin A, B, AB, AC, GDF8 and GDF11 protein reagents and the α-GDF11 antibody (MAB19581) were obtained from R&D Systems (Minneapolis, MN). ActRIIB.hFc was made at Regeneron.

**Surface plasmon resonance.** The affinities of REGN2477 and ActRIIB.hFc for human GDF8, GDF11, activin A, activin B, activin AB and activin AC were measured in surface plasmon resonance biacore experiments[7]. Briefly, GDF8, GDF11, activin A, activin B, activin AB and activin AC proteins were prepared at concentrations between 40 and 0.78 nM, and injected over captured REGN2477 or ActRIIB.hFc surface on a Biacore T200 (GE Healthcare, Pittsburgh, PA) at 25 °C. The affinity of the α-GDF11 antibody from R&D Systems (as described below) for GDF8 and GDF11 was also measured using the same protocol.

**Serum assays.** Activin A, GDF8 and GDF11 levels in serum from SCID mice (males, $n = 8$), rats (unknown gender, $n = 8$), cynomolgus monkeys (mixed gender, $n = 6$) and humans (mixed gender, $n = 10$) were determined using ELISA (activin A: Cat#DAC00B, GDF8: Cat#DGDF80) from R&D systems. GDF11 ELISA was adapted at Regeneron using the protocol published elsewhere[20]. To this end, we used high bind 96-well MESO Quickplex Plates (Mesoscale Discovery; #L55XB-3) coated with 50 μl per well of 2 μg ml$^{-1}$ (in PBS) mouse anti-human GDF11 (clone 743833, R&D #MAB19581) overnight at 4 °C with shaking. After washing with PBS supplemented with 0.05% Tween, 300 μl of 3% BSA/PBS was added to each well and the plate was incubated for 2 h at room temperature. A twelve-point calibration curve was prepared using threefold dilutions, starting with a prepared sample of 100 ng ml$^{-1}$ recombinant human GDF11 (R&D Systems #1958-GD-010) in 1% BSA/PBS. Samples were diluted 1:2 in 1% BSA/PBS. A total of 50 μl of sample or calibrator was added to each well and incubated for 3 h at room temperature. Following wash, 50 μl of SulfoTag labelled mouse anti-human GDF11 (labelled using MSD Gold Sulfo-Tag NHS-Ester, Mesoscale Discovery #R31AA-1) diluted to 4 μg ml$^{-1}$ in 1% BSA/PBS was add to each well and incubated for 2 h at room temperature with shaking. Plate was washed for a final time and 150 μl of 2× MSD Read Buffer T was added to each well. Plate was read on MESO QuickPlex SQ 120 reader. Sample concentrations were determined using a four-parameter-logistic equation. Human and monkey sera were obtained from BioreclamationIVT. All samples were assayed in duplicate. Plasma glucagon and insulin levels were determined using glucagon and mouse insulin ELISA (Mercodia). Plasma corticosterone levels were measured using a mouse corticosterone ELISA (Kamiya Biomedical Co., #KT-510).

**Pancreas histology.** Pancreata were weighed, fixed in 10% neutral buffered formalin solution for 48 h, and then embedded in paraffin. Two sections of the pancreas from each animal were stained with an anti-glucagon (REGN745, an anti-glucagon monoclonal antibody generated at Regeneron) or an anti-insulin (Dako) antibody and areas of glucagon and insulin positive cells were measured using Halo digital imaging analysis software (Indica Labs). The per cent of glucagon and insulin positive areas in proportion to the whole pancreas area were calculated. Cell mass was determined by multiplying the alpha cell or beta cell area for each animal against their pancreas weight.

**Mouse hypertrophy studies.** Male mice were housed 4–5 per cage on a 12 h light/dark cycle with free access to food (Purina Laboratory Rodent Diet 5001, LabDiet) and water. C.B.-17 SCID mice or C57BL/6 mice (all male; 9–10 weeks of age) were obtained from Taconic. All animal studies were conducted with permission and in accordance with the Regeneron Pharmaceuticals Institutional Animal Care and Use Committee.

For muscle hypertrophy studies, 9–10-week-old mice were randomized based on body weight with no blinding to groups of five or six animals and were dosed with control antibody, REGN2477, REGN1033, a combination of REGN2477 and REGN1033 or ActRIIB.hFc at the doses indicated in the text. A separate study in SCID mice was used to test the triple combination of REGN1033, REGN2477 and α-GDF11 blocking antibody. No statistical method was used to predetermine

sample size, which was estimated based on previously published studies[7]. For all SCID mouse studies, antibodies were administered in saline vehicle by s.c. injection on days 0, 3, 7 and 14 (4 doses total). For all C57BL/6 studies, mice were injected twice per week (six doses total) on days 0, 3, 7, 10, 14 and 17. All mice were euthanized on day 21, and TA and GA muscles were removed and weighed. Change in muscle weight (Δweight) was calculated as the raw difference in milligrams (mg) of each muscle from the average mg weight of the control group. No animals were excluded from the studies except as detailed below.

**Hindlimb suspension studies.** Recovery from HLS-induced muscle atrophy was assessed in 10-week-old C57BL/6 male mice using special cages for tail suspension to prevent hind limb weight bearing (Techshot Inc.) with free access to food and water. One group of mice was left unperturbed to serve as a negative control. After 7 days of suspension, the negative controls and one group of suspended animals were taken down and their TA and GA muscles were weighed. The remaining groups of mice were removed from HLS and randomized. Mice were treated with $25\,mg\,kg^{-1}$ of control antibody, REGN1033, REGN1033 and REGN2477 combination or ActRIIB.hFc by subcutaneous injection on the day of HLS removal (Day 7 of the study) and 3 days later (Day 10 of the study). At the end of the 7-day treatment period, TA and GA muscles were collected and weighed.

**Dexamethasone muscle atrophy studies.** Prevention of dexamethasone-induced atrophy was assessed in 9-week-old male C57BL/6 mice, grouped by body weight at the start of the study. Dexamethasone (Dexaject SP solution; Butler Schein) was administered at a rate of $23.0\,\mu g$ per day by micro-osmotic pump (Durect) for 2 weeks. During dexamethasone administration, four groups of mice ($n = 10$ per group) were subcutaneously injected with REGN1033, a combination of REGN1033 and REGN2477, ActRIIB.hFc or control antibody ($25\,mg\,kg^{-1}$). Test article administration started on the day of the pump implantation (day 0) and the mice received a total of four injections (days 0, 3, 7 and 10). A separate group ($n = 10$) was implanted with osmotic pumps delivering saline and administered control antibody ($25\,mg\,kg^{-1}$) at the time indicated above. At the end of 2 weeks of treatment, mice were weighed and euthanized. Both right and left TA and GA muscles were excised and weighed.

**Glucose homeostasis measurements.** Oral glucose tolerance test was performed as described[63] following 7 days of antibody treatment in 9-week-old male C57BL/6 mice ($n = 6$ per group) treated with either control antibody, a combination of REGN2477 and REGN1033 and ActRIIB.hFc at $10\,mg\,kg^{-1}$ each. All groups were fasted overnight and the next morning given an oral gavage of $2\,g\,kg^{-1}$ glucose at T0. Glucose was sampled by tail vein using an Accu-Chek Compact Plus Glucometer (Roche Diagnostics) at baseline, 15, 30, 60 and 120 min after administration.

**Spleen histology.** After harvest and weighing, spleens were placed lengthwise in a cassette and fixed in 10% formalin for 48 h, then transferred to 70% ethanol. Spleens were embedded in paraffin, sectioned and stained with H&E. Sections were photographed using a Zeiss Axio Imager with an Axiocam 503 colour camera at $\times 10$ and $\times 40$ magnification.

***Ex vivo* muscle physiology.** To determine whether combination treatment of REGN2477 and REGN1033 affected TA muscle function, *ex vivo* force was measured. Briefly, mice were anesthetized using 4.5% isoflurane and the right TA muscle was removed and placed in an oxygenated bath containing Krebs solution at 27 °C containing 10 mM glucose. One end of the TA muscle (with the femoral head) was secured to a submerged stanchion in the bath while the distal tendon was tied to the arm of a 305C Muscle Lever System (Aurora Scientific). Maximal twitch force was achieved by increasing the muscle length in small increments and stimulating with 1 Hz until no further force increase could be achieved. After a 5-min rest period, twitch force was determined by averaging the force from three consecutive 1-Hz stimulations, with 1 min of rest between stimulations. Peak isometric tetanic force was determined using a range of stimulations (40–150 Hz) with 2 min intervals between stimulations. Four mice across groups were excluded due to the bone or muscle splitting during setup or analysis. Data from these mice were only excluded from the physiology analysis; muscle weights and histology data were still utilized.

**Muscle fibre cross-sectional area and number.** After harvest and weighing, TA muscles were embedded in OCT and frozen in liquid nitrogen. The frozen muscles were sectioned perpendicular to their length and fibres were stained for laminin with a polyclonal rabbit anti-LAMA1 antibody (1:2,000, Sigma-Aldrich). Slides were scanned by Aperio Scanscope AT (Leica Microsystems Inc.). Fibre number and cross-sectional area were analysed by HALO software (Indica Labs). Fibre size is represented as average size of the total number of fibres in a cross-section of TA muscle.

Monkey muscle specimens were fixed in 10% neutral buffered formalin and transferred within 48 h to 70% ethanol. After fixation, the muscles were trimmed to obtain a maximum number of fibres in cross-section. Trimmed tissues were processed into paraffin blocks, sectioned at 4–6 μm, mounted on glass microscope slides and stained for dystrophin to outline the sarcolemma and allow for automated image analysis. Slides were scanned using the Hamamatsu NanoZoomer whole slide scanner and images were imported into the Visiopharm software platform. Image analysis protocols were created to count individual muscle fibres and the cross-sectional area was determined for each fibre using automated image analysis[64].

**Transcriptome sequencing and data analysis.** Protocols for our RNA-seq and the differential gene expression analysis have been previously described[65]. To summarize here briefly, we extracted (MagMAX kit, Life Tech) and purified mRNA (DynabeadsmRNA Purification Kit, Invitrogen) from the mouse TA muscle samples. Strand-specific RNA-seq libraries were constructed using ScriptSeq mRNA-Seq Library Preparation Kit (Epicentre) and amplified by 12-cycle PCR. Multiplexed single-read sequencing runs with 33 cycles were performed on Illumina HiSeq2000. The output BCL files were converted to FASTQ format via Illumina Casava 1.8.2. The reads were mapped to the mm10 mouse genome with one allowed mismatch using ArrayStudio (OmicSoft). Sense-strand exon reads were used to quantify the gene expression level. Genes flagged as detectable with an empirical minimum reads per kilobase per million (RPKM) of 0.1. For comparison between two groups of samples, genes were eliminated if they were not detectable in the number of samples that is greater or equal to the smaller sample size of the two comparing groups. For each gene, the fold change was calculated as the ratio of the arithmetic mean of the RPKM values between the two groups, with the statistical significance accessed by student *t*-test. Genes with a fold change $\geq 1.5$ in either direction and a *P* value $\leq 0.05$ were considered significantly differentially expressed. Pathway analysis was conducted with Ingenuity Pathway Analysis tool (QIAGEN). The transcriptome data have been deposited to ArrayExpression under accession number E-MTAB-4943.

**Cynomolgus monkey studies.** Two monkey studies using the combination of REGN1033 and REGN2477 were performed by two different study centres. Each institution had individual IACUC committees who reviewed and approved the study beforehand. Charles River Laboratories (Senneville, Canada), conducted the first study in both male and female cynomolgus monkeys ($n = 5$ of each gender per group) of 2–3 years of age, selected by the facility without blinding to the treatment groups, although the test facility was unaware of the actual targets of the individual compounds. Monkeys were weekly dosed intravenously (i.v.) for 5 weeks with REGN2477 or REGN1033 alone or in combination at the doses indicated in the text. Saline was used in the control groups. Whole body X-ray densitometry (DXA) was used to measure body composition using a Hologic Discovery A bone densitometer. DXA measurements (duplicate scans) were assessed once during the pretreatment phase and once towards the end of the dosing period.

The second study was conducted at SNBL (Everett, WA), again in both male and female cynomolgus monkeys ($n = 3$ of each gender per group) of 4–5 years of age. They were dosed i.v. weekly for 11 weeks with combination of REGN1033 and REGN2477 or ActRIIB.hFc at the doses indicated in the text. As in the previous study, test articles were given with no blinding to the treatment groups, although the test facility was unaware of the actual targets of the individual compounds. A third group was given saline as control. Whole body DXA was conducted twice during the pretreatment period and once during weeks 4, 8 and 11 of the dosing period using an X-ray densitometer (Hologic QDR-4500). Animals were sedated by ketamine and xylazine for the scans. At the end of the study, biceps were dissected and weighed and 0.5–1.5 cm³ of samples from the right biceps brachii was collected and sent to Wil Research (Hillsborough, NC) for morphometry analysis. Two control bicep weights and one ActRIIB.hFc bicep weight were excluded due to incomplete dissection, though these were still used for morphometry.

**Statistical analysis.** Data are presented as mean ± s.e. and values of $P < 0.05$ were considered statistically significant. Statistical significance was measured through unpaired, two-tailed Student's *t*-test for comparisons between two groups, one-way or two-way ANOVA with Bonferroni's *post hoc* analysis for studies with groups of three or more or two-way repeated measures ANOVA with Bonferroni's *post hoc* analysis for studies where time was a factor using Prism software (GraphPad Software). Alpha was set at $P < 0.05$. Data met the normality assumptions of the statistical tests.

**Data availability.** The transcriptome data have been deposited to ArrayExpression under accession number E-MTAB-4943. The data that support the findings of this study are available from the corresponding author upon reasonable request.

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

## Acknowledgements

We would like to thank Samantha Intriligator for help with the manuscript.

## Author contributions

E.L. and J.M. are joint first authors who contributed equally to this work. E.L., J.M., J.T., H.O., T.S., A.J.M. and J.G. designed the studies. J.M., L.M., J.P., K.C., A.P., T.W., E.N., A.R. and J.H.K. conducted the studies. E.L., J.M., W.F., L.M., J.T., J.P., H.O., K.C., E.N., A.P., T.W., Y.B., J.H.K., A.R., S.J. and T.S. analysed the data. E.L., J.M., A.J.M., G.D.Y. and J.G. wrote the manuscript.

## Additional information

**Competing interests:** All authors except E.L. and T.W. are current employees of Regeneron Pharmaceuticals. E.L. and T.W. were employees of Regeneron Pharmaceuticals when this study was conducted. J.M., W.F., L.M., J.T., J.P., H.O., K.C., E.N., A.P., Y.B., J.H.K., A.R., S.J., T.S., A.J.M., G.D.Y. and J.G. are currently employees and hold stock or stock options of Regeneron Pharmaceuticals. Regeneron Pharmaceuticals and E.L., T.S., A.J.M., G.D.Y. and J.G. hold patents for antibodies inhibiting GDF8 and activin A and uses thereof.

