## [Peer review file · Nature Communications]

REVIEWERS' COMMENTS:

Reviewer #1 (Remarks to the Author):

The revised manuscript from Latres, Mastaitis et al. contains a significant amount of new data addressing several questions raised during the initial review. The additional data demonstrating the safety of the antibody are more convincing. Furthermore, the data on using the antibodies to treat atrophy offers a new potent therapeutic avenue for a common problem.

1) An additional experiment that would further strengthen the paper would be to treat Mstn^{-/-} mice with REGN2477 to demonstrate that the observed findings were not due to an unexpected secondary effect between the REGN2477 and REGN1033 antibodies. Although I do believe the paper strong enough without this experiment, it would help..

2) In the abstract, it is unclear as to why the authors state that Activin A has a more prominent role in regulating muscle mass in primates compared to in mice. Is it simply because Activin A serum levels are higher in primates than in rodents? If Activin A was indeed more important than myostatin in regulating the muscle mass of primates, one would likely expect to see an overt effect following treatment with REGN2477 alone compared to treatment with REGN1033 alone. In fact, a significant increase in the lean mass of primates was only observed following treatment with REGN2477 in combination with REGN1033. This suggests that Activin A inhibition is only effective when myostatin is inhibited at the same time. Therefore, why such importance is attributed to one protein over the other is unclear. To this end, the title should be reworked to better reflect the major findings of the manuscript.

Reviewer #2 (Remarks to the Author):

The manuscript has been substantially improved.

Manuscript #: NCOMMS-16-21427B

Revised manuscript title: Activin A more prominently regulates muscle mass in primates than does myostatin

Reviewer #1:

1) An additional experiment that would further strengthen the paper would be to treat *Mstn*^{-/-} mice with REGN2477 to demonstrate that the observed findings were not due to an unexpected secondary effect between the REGN2477 and REGN1033 antibodies. Although I do believe the paper strong enough without this experiment, it would help..

We agree this would be a very interesting experiment, but it would take considerable time and effort to rederive our *Mstn*^{-/-} mice and, unfortunately, we would not be able to conduct it in a reasonable time frame.

2) In the abstract, it is unclear as to why the authors state that Activin A has a more prominent role in regulating muscle mass in primates compared to in mice. Is it simply because Activin A serum levels are higher in primates than in rodents? If Activin A was indeed more important than myostatin in regulating the muscle mass of primates, one would likely expect to see an overt effect following treatment with REGN2477 alone compared to treatment with REGN1033 alone. In fact, a significant increase in the lean mass of primates was only observed following treatment with REGN2477 in combination with REGN1033. This suggests that Activin A inhibition is only effective when myostatin is inhibited at the same time. Therefore, why such importance is attributed to one protein over the other is unclear. To this end, the title should be reworked to better reflect the major findings of the manuscript.

We changed the title of our manuscript to “Activin A more prominently regulates muscle mass in primates than does GDF8,” in accordance with the reviewer’s previous suggestion. And we agree with this title as our data shows comparable increase in muscle mass in primates following inhibition of either ligand (Fig. 5A), which is very different that observed in mice where GDF8 is clearly the dominant ligand. Therefore, we do feel the title is appropriate.

Reviewer #2 (Remarks to the Author):

The manuscript has been substantially improved.

We thank the author for their comment.